# Innovative Low-Cost Carbon/ZnO Hybrid Materials with Enhanced Photocatalytic Activity towards Organic Pollutant Dyes’ Removal

**DOI:** 10.3390/nano10091873

**Published:** 2020-09-18

**Authors:** Petronela Pascariu, Niculae Olaru, Aurelian Rotaru, Anton Airinei

**Affiliations:** 1“Petru Poni” Institute of Macromolecular Chemistry, 41A Grigore Ghica Voda Alley, 700487 Iasi, Romania; nolaru@icmpp.ro (N.O.); airineia@icmpp.ro (A.A.); 2Faculty of Electrical Engineering and Computer Science & MANSiD Research Center, Stefan cel Mare University, 13 Str. Universitatii, 720229 Suceava, Romania; aurelian.rotaru@usm.ro

**Keywords:** carbon/ZnO nanostructures, electrospinning, photocatalyst, photocatalytic activity

## Abstract

A new type of material based on carbon/ZnO nanostructures that possesses both adsorption and photocatalytic properties was obtained in three stages: cellulose acetate butyrate (CAB) microfiber mats prepared by the electrospinning method, ZnO nanostructures growth by dipping and hydrothermal methods, and finally thermal calcination at 600 °C in N_2_ for 30 min. X-ray diffraction (XRD) confirmed the structural characteristics. It was found that ZnO possesses a hexagonal wurtzite crystalline structure. The ZnO nanocrystals with star-like and nanorod shapes were evidenced by scanning electron microscopy (SEM) measurements. A significant decrease in *E*_g_ value was found for carbon/ZnO hybrid materials (2.51 eV) as compared to ZnO nanostructures (3.21 eV). The photocatalytic activity was evaluated by studying the degradation of three dyes, Methylene Blue (MB), Rhodamine B (RhB) and Congo Red (CR) under visible-light irradiation. Therefore, the maximum color removal efficiency (both adsorption and photocatalytic processes) was: 97.97% of MB (*C*_0_ = 10 mg/L), 98.34% of RhB (*C*_0_ = 5 mg/L), and 91.93% of CR (*C*_0_ = 10 mg/L). Moreover, the value of the rate constant (*k*) was found to be 0.29 × 10^−2^ min^−1^. The novelty of this study relies on obtaining new photocatalysts based on carbon/ZnO using cheap and accessible raw materials, and low-cost preparation techniques.

## 1. Introduction

A major worldwide problem of modern society is the disposal and treatment of wastewater coming from industrial processes. It is known that about 97% of water is represented by oceans in the form of salty water. This is not appropriate for human consumption or agricultural use, and only less than 3% of water is useful [1]. The quality and quantity of water are the main issues that need to be addressed by finding methods to eliminate contaminants or pollutants, which induce adverse environmental effects, as well as for human health. In addition, the residual liquids containing dyes coming from the textile industry often create severe environmental hazards because of their direct disposal into nearby water bodies. More than 15% of the dyes are lost in wastewater during dyeing operations. This affects the surface esthetic merit of water and reduces light penetration, disturbing aquatic life and hindering photosynthesis [2]. Furthermore, some dyes are either toxic, mutagenic or/and carcinogenic [1].

It is known that ZnO is considered one of the most important oxide semiconductors with a band gap energy of 3–3.37 eV and a large exciton binding energy of 60 meV, having a high capacity to decompose organic pollutants under ultraviolet (UV) irradiation or sunlight exposure [3]. Due to its unique properties, ZnO is widely used for a large variety of applications such as light-emitting diodes, nanolasers, piezo-electric devices, UV-shielding materials, antibacterial agents, field effect transistors, solar cells and gas sensors [4,5,6,7,8,9]. Moreover, this semiconductor material is considered an excellent photocatalyst for the degradation of some organic dyes in wastewater. Many researchers have been trying to improve the photocatalytic properties of ZnO by doping with various metals (La, Sm, Er, Ce, N, Ag, and so on,) [10,11,12,13], combining with other metal oxides (NiO, CeO_2_, SnO_2_, CuO, CdO, BaTiO_3_, NaNbO_3_, TiO_2_, Bi_2_O_3_, CuFe_2_O_4_) [14] or with various carbon-based nanostructures (multi-walled carbon nanotubes (MWCNTs), graphene, graphene oxide) [15,16,17,18].

Recently, composite materials based on a combination of metal oxide semiconductor nanomaterials and different types of carbon species have been intensively used as photocatalysts due to their remarkable physico-chemical properties and potential applications in water purification and environmental protection [1,19,20,21]. In addition, it was demonstrated that the development of materials based on ZnO and carbon leads to an increase in the stability and efficiency of photocatalytic performance [1,19,20,21]. One of the simplest and cheapest methods of obtaining carbon-based materials is the use of polymer matrices followed by their carbonization at high temperature in N_2_ atmosphere. The most widely used polymer in obtaining ZnO/carbon-based nanomaterials is polyacrylonitrile (PAN) and is generally used as electrodes for supercapacitors [22,23,24]. It was shown that carbon nanofibers play an important role in energy conversion and storage, catalysis, sensors, adsorption/separation, and biomedical applications due to its good conductivity and chemical stability, tunable structural flexibility, and low cost [24,25].

The main goal of this study is to point out the remarkable results of carbon/ZnO-based catalysts in photocatalytic degradation, starting from easily accessible and low-cost materials. For this purpose, electrospun fiber mats of cellulose acetate butyrate (CAB) were chosen for growing on them the desired ZnO nanostructures followed by their calcination at 600 °C in N_2_ atmosphere for 30 min. It is known that CAB is a thermoplastic polymer that softens in the first phases and then follows the degradation process. Moreover, it is a relatively inexpensive and accessible polymer. In this work, we aimed to produce a new type of material based on carbon/ZnO nanostructures that possess both adsorption and photocatalytic properties. To our knowledge, the development of carbon/ZnO nanostructures starting from CAB fiber mats obtained by electrospinning method, followed by ZnO nanostructures growth on them by dipping or hydrothermal method, and finally thermal calcination at 600 °C in N_2_ atmosphere for 30 min, and then their testing for adsorption/degradation of organic dyes, have not been reported in the literature so far. The details on the structural, morphological, and optical properties of the carbon/ZnO nanostructures were achieved and discussed. Furthermore, the development of new hybrid materials based on carbon/ZnO nanostructures will add valuable insights to scientific research by combining adsorption and photocatalytic processes.

## 2. Materials and Methods

### 2.1. Materials

Cellulose acetate butyrate was obtained by commercial sources, Eastman product—CAB 551-0.2—with 52 wt% butyryl content and 2 wt% acetyl content (Mn = 30,000), zinc acetate [Zn(CH_3_COO)_2_·2H_2_O], purchased from Sigma-Aldrich (Taufkirchen, Germany), and ammonia (NH_3_), purchased from Chemical Company SA, Iasi, Romania. All the chemicals were of reagent grade and were used without further purification.

### 2.2. Carbon/ZnO Hybrid Nanostructures Preparation

CAB fibers were prepared using the electrospinning method described in detail in previous work [26]. Briefly, the prepared viscous solution of 32% CAB in 2-methoxyethanol as the solvent was transferred in a syringe of the electrospinning setup. The main parameters of the electrospinning process were: high voltage source (25 kV), a 15 cm distance between the needle tip and the collector, and the flow-rate of 0.75 mL/h. Two methods were used in the growth of ZnO nanocrystals in CAB membranes: dipping and hydrothermal, followed by heat treatment at 600 °C for 30 min in N_2_ atmosphere to obtain carbon/ZnO hybrid nanostructures. The **M1** sample was obtained using the dipping procedure which consists in successive dippings of the membrane in an ammonium zincate bath with 0.1 M concentration and pH = 11, at room temperature, and then in a hot water bath, at about 97 °C, in 50 repeating cycles. After that, the sample **M1** was thermally treated at 240 °C in the air for 1 h.

The **M2** sample was prepared by the hydrothermal method consisting of: (i) ZnO seeded onto CAB nanofiber mat by 10 dippings; (ii) growth of ZnO nanocrystals by a hydrothermal method in ammonium zincate bath at (96–98 °C) for 3 h, followed by heat treatment at 240 °C in the air for 1 h. The carbon/ZnO hybrid nanostructures **M1 (T)** and **M2 (T)** were developed after calcining of membrane **M1** and **M2** at 600 °C in N_2_ atmosphere for 30 min. A representative diagram in preparing the carbon/ZnO hybrid materials is given in Scheme 1.

### 2.3. Characterization of Materials

X-ray diffraction (XRD) analysis of carbon/ZnO hybrid nanostructures as made on a Shimadzu Lab X XRD-6000 diffractometer (Columbia, United States) with CuK_α_ radiation (*λ* = 0.15418 nm). The morphological properties of the obtained materials were demonstrated by scanning electron microscopy (SEM), using JEOL JSM 6362LV electron microscope (Japan). A Bruker Fourier transform infrared (FTIR) spectrometer (VERTEX 70, Ettlingen, Germany) equipped with a Deuterated Lanthanum α Alanine doped TriGlycine Sulphate (DLaTGS) detector was used for the analysis of the FTIR spectra of materials. Diffuse reflectance of carbon/ZnO hybrid materials was performed by ultraviolet–visible (UV–Vis) reflectance spectra measured on an Analytik Jena UV-Vis 210 spectrometer (Jena, Germany). Then, the band gap values were obtained using Kubelka–Munk function (KM) and by plotting [*F*(*R*_∞_)*hν*]^2^ vs. *hν*.

### 2.4. Photocatalytic Tests

The adsorption and photocatalytic efficiency of carbon/ZnO hybrid nanostructures have been evaluated by degradation of Methylene Blue (MB), Congo Red (CR), and Rhodamine B (RhB) dye in aqueous solutions under visible light irradiation (100 W tungsten lamp source). More details on the degradation procedure and working conditions have been reported previously [27]. Initially, 5 mg of each material were dispersed in 10 mL of dye solution with an initial concentration of 10 mg/L MB, CR, and 5 mg/L of RhB, respectively. Then, the solutions were stirred in the dark for 2 h to establish an adsorption-desorption equilibrium. The photocatalytic activity of the carbon/ZnO hybrid nanostructures was investigated by photodegradation of MB, CR, and RhB dyes using the same experimental setup and degradation procedure as reported by the authors in a previous work [28]. The UV–Vis absorption profiles for the initial dye solution and after exposure to visible light at various time intervals were obtained using UV-Vis spectrophotometer (SPECORD 210Plus, Analytik Jena, (Jena, Germany). Adsorption capacity (Q_e_, mg/g) and removal efficiency (%) for adsorption and degradation of MB were calculated using the following equations [29]:(1)$$q_{e}=\frac{\left(C_{0}-C_{e}\right)\times V}{m}\times 100,$$
(2)$$\mathrm{Color}\;\mathrm{removal}\;\mathrm{efficiency}\;\left(\%\right)=\frac{C_{0-}C_{e}}{C_{0}}\times 100,$$
where *C*_0_ is the initial MB concentration (mg/L) and *C_e_* is the MB concentration at the time *t* (mg/L), *m* is the catalyst mass (g), and *V* is the solution volume (L).

## 3. Results

### 3.1. X-ray Diffraction (XRD) Characterization

X-ray diffraction (XRD) patterns of **M1 (T)** and **M2 (T)** materials are shown in Figure 1 and confirm the crystalline phase of ZnO with the hexagonal wurtzite structure.

The peaks corresponding to this structure are found at 2θ of 31.78° (100), 34.46° (002), 36.26° (101), 47.64° (102), 56.62° (110), 62.90° (103), and 67.10° (112) and belong to pure ZnO structure Joint Committee on Powder Diffraction Standards (JCPDS No. 89-1397). The main parameters that can be deduced from the analysis of X-ray diffractograms are summarized in Table 1, and for their calculation, the diffraction peaks corresponding to the Miller indices (100), (002) and (101) were used. In addition, to estimate the crystallite size (D), the spacing distance between crystallographic planes (d_hkl_), the lattice parameters *a* and *c*, the Zn–O bond length (L) and the microstrain (ε), the authors utilized the equations described in detail in previous work [11].

From the analysis of the lattice parameters *a* and *c* presented in Table 1, it can be seen that they do not show significant changes after the carbonization of the organic material, which confirms that the hexagonal wurtzite structure of ZnO is maintained. Besides, the ratio *c*/*a* is practically constant, which indicates that the hexagonal wurtzite structure of ZnO structure does not change. Significant changes can be observed for the crystallite size (D) and the microstrain (ε) parameter. The crystallite size values vary between 22.88 nm (corresponding to **M1 (T)** sample) and 36.13 nm (registered for **M2 (T)**), respectively. In addition, a discreet broadening of **M1 (T)** signals was observed which may be ascribed to the presence of a star-like shape of the ZnO crystallites, having a more multidirectional distribution. Moreover, it is well known that a smaller size of crystallites will induce a broadening of the signal. The microstrain (ε) parameter increases from 0.353 corresponding to sample **M2 (T)** to 0.564 for sample **M1 (T)**, probably due to the shape change of the nanostructures and the carbon content of the samples. XRD analysis (Figure 1 (inset)) suggests the presence of carbon in both samples with broad diffraction peaks between 20° and 30°, which was assigned to the (002) lattice planes in the graphitic structure [20]. A significant difference can be observed in the value obtained for crystallites size in XRD compared to those observed in SEM. It is known that the formation of these nanostructures (star-like and nanorod shapes in our case) takes place in two stages: nucleation and growth. In the first stage, small nuclei are formed which, as the reaction progresses, these nuclei grow further to produce star-like and nanorod ZnO crystallites, which are the building blocks for the crystals observed in SEM images [30,31,32,33,34].

### 3.2. Fourier Transform Infrared (FTIR) Analysis

Figure 2 shows the FTIR spectra of CAB nanofibres and carbon/ZnO corresponding to **M1 (T)** and **M2 (T)** nanostructured materials registered between 370 and 4000 cm^−1^.

It is known that ZnO has an intense broad band between 420 cm^−1^ and 510 cm^−^^1^ due to two transverse optical stretching modes of ZnO [35,36]. In our case, two characteristic absorption bands located at 397 cm^−^^1^ and 497 cm^−^^1^ were observed corresponding to **M2 (T)** material, as well as an absorption band at 424 cm^−^^1^ of **M1 (T)**, respectively. The occurrence of these two bands in the FTIR spectrum for sample **M2 (T)** it is due to the nanorod shape nanostructures. Wu et al., [37] state that the transition from 0D nanostructures (nanoparticles) to 1D (nanorod) leads to the appearance of two main absorption maxima in FTIR spectra in this range. The presence of vibration bands at the wavenumbers of 1614 cm^−1^ and 1529 cm^−1^ assigned to the asymmetric stretching vibration and symmetric stretching vibration of C=C bonds indicates the removal of functional groups and the successful carbonization of the new material, as it was confirmed by other authors for similar systems [38]. This aspect is very important since amorphous carbon is known as a very good adsorbent [39]. The absorption band located at 3425 cm^−1^ belongs to the stretching vibration of O–H groups due to the absorbed water on the surface of the carbon/ZnO materials. The bands around 1083 cm^−1^ are associated with bending vibrations of various ether bridges coming from the residual polymeric material.

### 3.3. Morphological Characterization

The SEM image of the CAB microfiber obtained immediately after the electrospinning process is shown in Figure 3.

This micrograph confirms the formation of a membrane with uniform and smooth microfibers having dimensions of 1 µm. After this, star-shaped crystals and nanorods were grown on these membranes by dipping and hydrothermal methods, followed by calcination at 240 °C in the air for 1 h. SEM images shown in Figure 4 for hybrid CAB/ZnO nanostructures obtained by the dipping method indicate a structure composed of ZnO nanocrystals with a star-like shape. It can be observed that the same structure was maintained after calcination at 600 °C in N_2_ atmosphere for 30 min for **M1 (T)** nanostructure (Figure 4).

The materials obtained by the hydrothermal method (**M2** and **M2 (T)**) show a nanorod type structure with an average diameter of about 700 nm and lengths around 5 µm according to the SEM images represented in Figure 5.

### 3.4. Optical Properties

The most important parameter that significantly influences the photodegradation process is represented by the energy band gap of materials. The value of this parameter was assessed by UV–Vis reflectance experiments, followed by applying the Kubelka–Munk equation (Equation (3)) and Tauc relation (Equation (4)) [40].
(3)$$F(R_{\infty })=\frac{{\left(1-R_{\infty }\right)}^{2}}{2R_{\infty }},$$
where *F*(*R*_∞_) is the so-called remission or Kubelka–Munk function and *R*_∞_ is the reflectance of the samples.
(4)$${\left[F\left(R_{\infty }\right)h\nu \right]}^{2}=A\left(h\nu -E_{g}\right),$$
where *A* is a constant, *E*_g_ is the optical band gap of the material.

The energy band gap values of carbon/ZnO nanostructures were obtained by plotting [*F*(*R*_∞_)*hν*)]^2^ versus *hν* and extrapolating the linear portion of the absorption edge to find the intercept with photon energy axis as shown in Figure 6.

A significant decrease in *E*_g_ value was observed for carbon/ZnO hybrid materials. Thus, for **M1 (T)** the *E*_g_ was found to be 2.51 eV, while for **M2 (T)** 2.73 eV, respectively. These values are lower compared to those obtained for CAB/ZnO (3.21 eV and 3.31 eV) nanostructures reported in our previous works [26]. This decrease of *E*_g_ could be ascribed to the enhanced conductivity, confirmed by other authors for similar systems [41]. It can be seen that the presence of carbon in ZnO nanostructures leads to a change in the electronic energy levels. For example, similar results were obtained for hybrid RGO-ZnO where *E*_g_ decreases to 2.16 eV as compared to pure ZnO (3.06 eV) [42]. Another study reported by Rahimi et al., [15] showed that the *E*_g_ value decreases from 3.2 eV (ZnO) to 2.8 eV for ZnO nanorod/graphene quantum dot composites, respectively. The authors associate this phenomenon to the formation of Zn–O–C or Zn–C chemical bonds in the composites obtained.

### 3.5. Photoluminescence Study

The analysis of the photoluminescence properties is closely related to the photocatalytic properties of the developed catalysts and help us to understand the recombination processes of the photogenerated electron-hole pairs. Therefore, the emission spectra obtained under 300 nm and 320 nm excitation wavelengths are presented in Figure 7.

It can be seen that the emission spectra corresponding to sample **M1** (CAB/ZnO) show several emission bands at 327 nm, 391 nm, 421 nm, 444 nm, and 484 nm, respectively. The UV emission bands from 327 nm (Figure 7a) and 350 nm (Figure 7b) can be assigned the near band edge (NBE) emission, and may be due to free exciton recombination [43]. It is known that the emission bands in the visible spectrum are due to different intrinsic defects of ZnO nanostructures, which include oxygen vacancies (*V*_O_), zinc vacancies (*V*_Zn_), oxygen interstitials (*O*_i_), zinc interstitials (*Zn*_i_) and oxygen antisites (*O*_Zn_) [27].

The emission spectrum of carbon/ZnO sample provides weak photoluminescence compared to sample **M1**. This means that the absorbed light is used efficiently in generating hole-electron pairs, without losing in the form of photoluminescence. The band located in the blue region practically disappeared, and the bands at 327 (Figure 7a), 350 nm (Figure 7b) and 484 nm become very weak. According to other studies [15], this large decrease of photoluminescence of carbon/ZnO nanostructures may indicate a large decrease in the radiative recombination rate of electron-hole pairs.

### 3.6. Adsorption/Photocatalytic Properties

#### 3.6.1. Adsorption/Photocatalytic Properties of Carbon/ZnO Hybrid Nanostructures for Degradation of Organic Pollutants

In the first stage of this study, the degradation efficiency of rhodamine B (*C*_0_ = 5 mg/L) for the starting samples (**M1** and **M2**) and the calcined samples (**M1 (T)** and **M2 (T)**) in N_2_ was performed. The blank test (without catalyst) was initially evaluated after 4 h and showed that the intensity of the absorption band of RhB decreases slightly, yielding 1.39% in dye degradation. Figure 8 shows the evolution of the absorption spectra of all materials after adsorption for two hours to establish the adsorption/desorption equilibrium of dye on the photocatalyst surface, followed by the degradation between 4 and 20 h depending on the efficiency of the samples.

From the analysis of the samples, it was noticed that for the samples **M1** and **M2** the adsorption process was very small (5.56% for **M1** and 1.08% corresponding to **M2**). A significant increase in the adsorption process occurs after the carbonization of materials, yielding adsorption efficiency between 89.61% (**M1 (T)**) and 46.59% (**M2 (T)**), respectively. This increase was attributed to the inclusion of carbon in the newly developed hybrid materials. It is known that carbon-based materials lead to an increase in adsorption, conductivity, as well as a decrease in the energy band gap [25]. The most outstanding result, which cumulates both the adsorption/photocatalytic processes, was registered for **M1 (T)** with an efficiency of up to 98.34%.

In the next part of this study, the effect of the initial MB dye concentration on the **M1 (T)** nanostructure activity was investigated. To assess each contribution, adsorption and photodegradation, measurements for five initial dye concentrations (7, 10, 13, 17, and 20 mg/L) were performed. Figure 9 shows that the color removal efficiency in the adsorption process increased with the decrease in the initial dye concentration. Initially, it can suggest that this process is apparently significant, but after calculating, the adsorption constant *Q_e_* (mg/g) for all concentrations was the same (12–13 mg/g) for all samples. Under these conditions, in the next part of this work, the photodegradation of MB dye was evaluated without taking into account the adsorption process.

#### 3.6.2. Photocatalytic Activity of Carbon/ZnO Hybrid Nanostructures for Degradation of Methylene Blue (MB) Dye

Figure 10a,b show the evolution of the UV–Vis absorption spectra for MB dye degradation in presence of both catalysts (**M1 (T)** and **M2 (T)**) under visible light irradiation for 4 h (without previous the 2 h adsorption process).

It was observed that after 4 h of visible light irradiation the absorption band at 665 nm decreases to almost 0, reaching a maximum efficiency of 99.69% for sample **M1 (T)**. The **M2 (T)** sample reveals a slower decrease in the degradation efficiency, yielding 60.59%. We consider that this difference between the values of the photocatalytic degradation efficiency would be due to the different shapes and the presence of carbon in the nanostructures, giving a lower value of the band gap for the **M1 (T)** sample.

Quantitative estimation of degradation kinetics of MB dye was performed using a pseudo-first-order kinetics model according to the following equation: ln(*C*_0_/*C*_t_) = *kt*, *C*_0_ is concentration of dye solution before irradiation, *C_t_* is concentration of dye solution after t minutes of irradiation, and k is the pseudo-first-order rate constant. The value of the reaction constant for sample **M1 (T)** was calculated by plotting ln(*C*_0_/*C*_t_) versus irradiation time *t* (see Figure 10c) and was found to be 0.29 × 10^−2^ min^−1^ with the value *R*^2^ = 0.9884 attributed to a pseudo first-order reaction kinetics.

To demonstrate the adsorption/photocatalytic properties of the new carbon/ZnO hybrid nanostructures it was performed experiments in photocatalytic degradation of MB (initial concentration 10 mg/L), RhB (5 mg/L) and CR (10 mg/L) as a test reaction. Very good results were recorded for the degradation of all dyes tested with the following maximum color removal efficiency (both adsorption and adsorption + photocatalytic processes after 4 h of irradiation): 97.97% for MB (*C*_0_ = 10 mg/L), 98.34% for RhB (*C*_0_ = 5 mg/L), and 91.93% for CR (*C*_0_ = 10 mg/L), respectively (Figure 11).

Therefore, it can be stated that these materials could be employed as promising low-cost photocatalysts with impressive efficiency for potential applications in water purification and environmental protection. Under the given conditions—visible light irradiation at low power (a 100 W tungsten with a power of 102.74 kJ·m^−2^·h^−1^), a moderate amount of catalyst (0.5 g/L) and 4 h degradation process—the newly obtained materials present an outstanding response towards organic dyes degradation, with a removal efficiency of 91.93%, 97.97% and 98.34%, depending on the type of dye.

Table 2 reveals the photocatalytic activities represented by the values of the reaction rate constant *k* (min^−1^) or degradation efficiency (%) for the degradation of different dyes in the presence of ZnO/carbon-based catalysts. As can be seen, good results were found for these materials based on different carbon nanostructures (reduced graphene oxide, graphene quantum dot, graphene oxide, carbon nanofibers, carbon) [15,16,17,18,44]. All authors reported an improvement in photocatalytic activity for these composite materials as compared to ZnO. Instead, the materials analyzed in this study showed enhanced photocatalytic efficiency after 4 h under visible light irradiation at low intensity visible light in the degradation of all dyes (MB, RhB, and CR).

According to the above results, a mechanism has been proposed to explain the improvement of the photocatalytic efficiency of the carbon/ZnO nanostructures as compared to pure ZnO (Figure 12).

The degradation mechanism takes into account the cooperative or synergetic effects between the carbon generated during calcination and zinc oxide (Figure 12).

During photon excites, electron hole pairs are generated in the ZnO valence band. These excited electrons will move in the conduction band of ZnO and then diffuse toward the surface of the carbon particles. The holes excess in the valence band will migrate to the surface on ZnO, where they react with water molecules or hydroxyl ions to generate active species of hydroxyl radicals (OH). This method suggests that the photogenerated electrons and holes were effectively separated. Moreover, the good separation of the photogenerated electrons and holes in the carbon/ZnO nanostructures is supported by the photoluminescence investigations of ZnO and carbon/ZnO. According to Figure 7, carbon/ZnO nanostructures revealed weaker emission intensity compared to ZnO. This aspect suggests that the recombination of the photogenerated charge carrier was highly inhibited in the carbon/ZnO nanostructures. The efficient charge separation could induce the increase of the charge carriers’ lifetime by enhancing the efficiency of the interfacial charge transfer of the adsorbed substrates. This discussion is also supported by other studies regarding similar systems [17,18].

## 4. Conclusions

Carbon/ZnO nanostructures were obtained in three stages: CAB microfiber mats were prepared by the electrospinning method, ZnO nanostructures were grown by dipping and hydrothermal methods, followed by thermal calcination at 600 °C in N_2_ atmosphere for 30 min. XRD measurements of photocatalysts confirmed a hexagonal wurtzite crystalline structure of ZnO, as well as the presence of carbon with (002) lattice planes. SEM measurements showed the formation of nanostructures with star-like and nanorod shapes. The *E*_g_ value decreased significantly for carbon/ZnO hybrid materials (2.51 eV) as compared to ZnO nanostructures (3.21 eV). The photocatalytic efficiency for degradation of Methylene Blue (MB), Rhodamine B (RhB) and Congo Red (CR) dyes under visible-light irradiation has been improved as compared to ZnO. These new materials showed an improvement of the photocatalytic degradation efficiency for the RhB dye with approximately 80% as compared to the ZnO (control samples). The carbon/ZnO hybrid materials recorded a color removal efficiency (adsorption/photocatalytic process) between 91% and 98%, depending on the type of dye. All the experiments were performed under friendly environmental conditions: visible light irradiation at low power and a moderate amount of catalyst (0.5 g/L). Moreover, the value of the rate constant was found for this material to be 0.29 × 10^−2^ min^−1^. Therefore, the prepared carbon/ZnO materials from easily accessible and low-cost materials together with their impressive performance place them among photocatalysts for practical applications in wastewater purification.

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
