# Peer review of "Innovative Low-Cost Carbon/ZnO Hybrid Materials with Enhanced Photocatalytic Activity towards Organic Pollutant Dyes’ Removal"

_nanomaterials, 2020, doi:10.3390/nano10091873_

Round 1
Reviewer 1 Report
In Scheme 1, all photo (M1,M1(T),M2,M2(T)) should be placed
The difference of Method 1 and Method 2 is unclear.
Method 2 include the dipping process to H2O or not?
Method 1 and Method 2 must be displayed in the complete separated sequence.
In Figure 1, the graph of M1 and M2 should be exchanged.
Why the ZnO peak of M1(T) became broaden and weaker than those of M2(T)?
Author should discuss in text.
When the carbon peak is discussed, the all XRD pattern (CAB NFs, M1 and M2)
must be shown.
In Figure 2, the all FTIR spectra (CAB NFs, M1 and M2) must be shown.
Figure 6, the graph of (a) and (b) are not consistent.(the kink structure and the order of value around 3.6eV)
For Figure 6, the reflectance of (CAB NFs, M1 and M2) should be shown.
In Figure 7, the spectra of (CAB NFs) should be shown.
The diffinition of "color removal efficiency" should be described.
How to treat the adsorption time?
In Figure 8, 9,and 10, both data for RhB and MB must be shown.
Do not mix the result for RhB and MB dye.
In Figure 11, "color removal efficiency" must depend the value of C_0
Fig. 11(d) is meaningless.
In Table 2, the column of Reaction rate constant and degradation efficiency must be separated. Do not mix them
Actually, the comparison of degradation efficiency is meaningless, because of the dependency of the experimental condition(dye, concentration, light source, etc...).
Author must focus on only the reactoin rate constant k(min-1).
And Figure 12 is not supported by the result.
Actually, the ZnO structure are quite different in M1 and M2. There is no discussion in the text releted to Figure 12.
English corrections:
dye solutuon-> dye solution
easely accesible->easily accessible
Author Response
Thank you very much for your attentive and competent analysis of our paper.

Reviewer 2 Report
The paper presented the synthesis of low cost carbon/ZnO hybrid materials using electrospinning method and the application in organic pollutant dyes removal. The method and the result seem interesting but there are quite some flaws in the analysis and the interpret of the result.
- The authors used two methods to grow ZnO on the electrospinning prepared CAB fibers: dipping method and hydrothermal method (samples M1 and M2). Later the as-prepared ZnO/CAB samples were thermal treated in air and N2 (samples M1(T) and M2(T)), which result in four different samples. It is well known that thermal treatment at this temperature will not only change the composition of the samples (CAB carbonized) but also modify the crystallinity of ZnO. This was now well discussed in the paper.
- In line 151 and Table 1, it was mentioned that the crystallite size of ZnO was smaller than 40 nm, which does not agree with the SEM analysis in Figure 4 and 5. Especially in Figure 5, the ZnO seems to be nanorods with diameter closes to 1 micron and length 10 micron range.
- In Figure 1, there was no explanation about that the index refer to (e.g. diffraction from ZnO), besides some of the XRD peaks were not correctly assigned index (e.g. in each spectrum, two peaks were assigned to (002) and two were assigned to (110)).
- In Figure 6, the two panels were not named besides the data in the two panels do not correspond to each other—M1(T) in the left panel seems to correspond to M2(T) curve in the right panel.
- In Figure 7, photoluminescence spectra from M2 samples were not presented, the instrumental information was missing. The authors assigned the 327nm emission (ex 300nm) and 350nm emission (ex 320nm) to the near band emission (NBE) from ZnO, which are not totally correct. First the ZnO NBE usually located at around 370-380 nm, which usually does not change with the excitation wavelength. Second, the changes of ZnO NBE emission in the cited reference (29) was due to doping of Ce, which is not related with the situation in this case.
- Sample M1(T) clearly shows better photocatalytic performance than that of sample M2(T) in degradation of all the measured organic pollutants. The authors did not give the possible reason.
- Among all the 30 cited references, seven of them were from their own work. Many well-known review papers were not mentioned in the paper.
In my opinion, This paper needs major revision before it can be accepted for publication.
Author Response

(The authors gave the same response as above.)

Reviewer 3 Report
Pascariu and colleagues prepared carbon/ZnO hybrid materials using systematic approaches involving hydrothermal method. Many characterization techniques have been used to examine the morphology and properties of the as-prepared materials including XRD, SEM, FTIR, UV-vis and others. Furthermore, the authors studied the photocatalytic activity of the hybrid materials for the removal of organic pollutant dyes. I think this paper is interesting and could be accepted for publication only after carrying out some major revisions.
- Please shorten the length of Abstract. Currently, the Abstract is too long, providing too much detail of experimental section. The readers do not really want to see experimental detail (what type of instruments you used to characterize the samples), but they want to see what type of results were obtained and how important they are.
- There are too many grammatical errors throughout the manuscript. The authors should get help from someone who is Native or fluent in English to revise their manuscript.
- Citations (referencing) in this paper looks bit poor. Some of the citations have been made in places where no citation is needed. But there are some places citations must be made left empty (no reference).
For example, the authors should provide citation:
- "More than 15% of the dyes are lost in wastewater during dyeing operation. This affects the surface esthetic merit of water and reduces light penetration, disturbing aquatic lives, and hindering photosynthesis."
- "It is known that ZnO is considered one of the most important oxide semiconductors with a band gap energy of 3–3.37 eV and a large exciton binding energy of 60 meV, having a high capacity to decompose organic pollutants under UV irradiation or sunlight exposure."
- Due to its unique properties, ZnO is widely used for a large variety of applications such as light-emitting diodes, nanolasers, piezo-electric devices, UV-shielding materials, antibacterial agents, field effect transistors, solar cells and gas sensors.
Reference number 1 has been used for everywhere in the manuscript (Ceram. Int. 2019, 45, 11158–11173.). The authors should use more relevant references in some places.
- For the applications of ZnO, the authors should consider citing:
- Henk J. Bolink et al., Air stable hybrid organic-inorganic light emitting diodes using ZnO as the cathode, Appl. Phys. Lett. 2007, 91, 223501. (LED)
- Liao et al. Low threshold room-temperature UV surface plasmon polariton lasers with ZnO nanowires on single-crystal aluminum films with Al2O3 interlayers, RSC Advances, 2019, 9, 13600-13607. (Laser)
- Yoon et al., Development of Al foil-based sandwich-type ZnO piezoelectric nanogenerators, AIP Advances, 2020, 10, 045018. (piezoelectric device)
- Saha et al. Remarkable Stability Improvement of ZnO TFT with Al2O3 Gate Insulator by Yttrium Passivation with Spray Pyrolysis, Nanomaterials 2020, 10, 876. (Transistor)
- Gao et al., Synthesis of ultra-long hierarchical ZnO whiskers in a hydrothermal system for dye-sensitised solar cells, RSC Advances., 2016, 6, 109406 (solar cells)
- The authors wrote in Introduction: "Recently, the composite materials based on a combination of metal oxide semiconductor nanomaterials and different types of carbon species have been intensively used as photocatalysts due to their remarkable physico-chemical properties and potential applications in water purification and environmental protection."
Here again, reference 1 was cited. Please provide some more relevant citation in addition to the reference 1. Very important reviews in the field are as follows and should be cited:
- Batmunkh et al., Carbonaceous Dye‐Sensitized Solar Cell Photoelectrodes, 2015, 2, 1400025.
- Hu et al., A brief review of graphene–metal oxide composites synthesis and applications in photocatalysis, Journal of the Chinese Advanced Materials Society, 2013, 1, 21-39
- Concina et al., Semiconducting Metal Oxide Nanostructures for Water Splitting and Photovoltaics, Advanced Energy Materials, 2017, 7, 1700706.
- The authors should consider improving the Scheme 1. I think the abbreviation like (M1, M1 (T), M2 and M2 (T)) are not very clear.
- I am very impressed by the XRD patterns of the samples. Carbon particles can be clearly observed from the SEM images, but the XRD peak intensity for carbon is very weak. Can authors explain why?
- For Figure 4 and Figure 5, how about using a, b, c and d labelling? It may look better?
Author Response

(The authors gave the same response as above.)

Round 2
Reviewer 1 Report
previous referee comments noted that author should discuss xrd data, FTIR spectra, reflectance, betwenn M1, M2 and M1(T), M2(T).
But, I could not find it in the revised one.
it is difficult to discuss band structure based on Figure 6, because ZnO badngap is the same among all the samples, or not.
Author Response
Thank you for revision.

Reviewer 2 Report
The authors did not give the best answer to my questions considering photoluminescence spectra, but they made their point: the focus of is paper is photocatalytic performance of the carbon/ZnO hybrid materials. Besides, the authors' reply to other questions/comments is reasonable and they made careful revision on the manuscript. The quality of the paper is much improved.
There is a minor revision needed for the figures: for exampe, the two figures in figure 6 named as (a) and (b) in the figure caption, but they are not marked/labeled in the figures. Please check each figures and fix them.
I recommend this paper for publication after the revision.
Author Response
Thank you for the revision.

Reviewer 3 Report
The authors have made revision in their manuscript as required. The revised manuscript is satisfactory. However, the track changes in the manuscript has not been removed. The paper is acceptable, but the final version should be submitted without the track changes.
Author Response
Thank you for the revision.
Round 3
Reviewer 1 Report
In Figure 2, the vertical axis should not be "absorbance", (see, Fig.3 in ref App. Surf. Sci., 455 (2018), p.61-69)
In Fig. 6, in comparison with Fig 6. in ref App. Surf. Sci., 455 (2018), p.61-69, it is difficult to discuss band gap behavior because it looks bas structure like anymore.
Author Response
Thank you for your manuscript revision.
